# Multi-axial strain mapping to characterise structure and material properties of the human hip capsule

Kabelan J. Karunaseelan[1]*, K.C. Geoffrey Ng[2,3,4,5], Sarah K. Muirhead-Allwood[6], Richard J. van Arkel[1], Jonathan R. T. Jeffers[1]

1 Biomechanics Group, Mechanical Engineering Department, Imperial College London, London, United Kingdom, 2 Department of Medical Biophysics, Western University, London, Ontario, Canada, 3 Department of Medical Imaging, Western University, London, Ontario, Canada, 4 Department of Surgery, Western University, London, Ontario, Canada, 5 Robarts Research Institute, Western University, London, Ontario, Canada, 6 The London Hip Unit, London, United Kingdom

* k.karunaseelan18@imperial.ac.uk

## Abstract

The biological composition and spatial arrangement of the tissue's constituents are directly related to its condition and function. Conventional inspection techniques such as optical microscopy and exogenous staining have limited ability to capture the heterogeneity and anisotropy of biological tissues. Here, we apply a commercial 2D digital image correlation (DIC) system integrated with a biaxial testing machine to quantitatively characterise the fibrous structure of biological materials. The approach applies a homogeneous biaxial stress field across the tissue and uses optical measurement of the resulting strain field to identify load-bearing collagen architecture. Under this loading condition, collagen bundles deform less than the surrounding matrix, so low-strain bands and their principal directions indicate ligament locations and orientations. The method was validated using an artificial anisotropic material and ex vivo skin, and was then applied to the human hip joint capsule to demonstrate its ability to characterise complex collagen networks. Testing of nine excised hip capsule specimens revealed the collagenous network and the confluence between its fibrous structures. The locations and orientations of seven ligamentous regions were detected and matched to previously published anatomical descriptions. Using strain as a quantitative measure of ligament anatomy further enabled extraction of local mechanical properties, including the tangent modulus, across the entire tissue in a single test. By combining a biaxial testing machine with a commercial 2D DIC system, this study demonstrates a practical and scalable approach for quantifying tissue structure-function relationships across whole tissues.

provided the original author and source are credited.

**Data availability statement:** All relevant data are within the paper and its Supporting Information files.

**Funding:** Funding for this research was provided by the National Institute for Health Research (NIHR300013). Richard J. van Arkel is funded by UK Research & Innovation (EPSRC fellowship EP/Z536039/1).

**Competing interests:** The authors have declared that no competing interests exist.

## 1 Introduction

The biological composition and spatial arrangement of a tissue's constituents play an essential role in its mechanical behaviour and biological function [1]. In soft biological tissues such as ligaments, tendons, heart valves and skin, the extracellular matrix (ECM) consists of interwoven collagen fibres [2–5]. The arrangement of these fibres is configured in specific architectures to support various functional requirements of the native biological tissue [6]. A key biomedical challenge has been to examine the collagen fibre architecture of these biological tissues to elucidate the relationship between fibre structure and the tissue's mechanical function [7–10].

Previous studies have typically examined biological tissues in an unloaded state or by chemical fixation in a loaded state, limiting the ability to relate observed collagen architecture to the mechanical role of the fibres during loading [11,12]. Fibre recruitment occurs only when the tissue is placed under tension, so assessing collagen architecture while the tissue is loaded provides a more functionally relevant description of the fibrous network [13]. This perspective is important for interpreting how alterations in collagen structure contribute to diseases such as tendon injury [14], liver fibrosis [15], cerebral aneurysms [16], and stenosis in heart valve leaflets [5]. In this study, our aim is not to examine how the collagen network changes with different load levels, but rather to quantify its organisation when subjected to a controlled biaxial stress field.

Many imaging and characterisation techniques have been used to study the macroscopic and microscopic organisation of soft collagenous tissues such as small angle light scattering (SALS) [17], polarized spatial frequency domain Imaging (pSFDI) [18], contrast-enhanced microtomography (μCT) [19], polarised light imaging (PLI) [20], magnetic resonance imaging (MRI) [21], second harmonic generation (SHG) microscopy [22], confocal microscopy [23], histology [24,25], and digital volume correlation (DVC) [26]. Each method offers unique advantages and limitations for examining collagen fibre structure. For example, recent biaxial mechanical systems have integrated SALS, pSFDI, or PLI to characterise collagen fibre organisation and dynamic realignment of tissues [18,27–29]. These techniques provide relatively large fields of view (typically on the order of $\mu m - mm$) with high specificity and sensitivity to collagen; however, they can be expensive to implement [17,18]. Contrast-enhanced μCT improves soft-tissue visibility through staining, although achieving reliable contrast can be challenging and preparation times are often long [19]. SHG microscopy and histology allow visualisation of individual collagen fibres at high resolution, but the trade-off is a limited field of view, making whole-tissue assessment impractical [1,22,30]. Non-optical techniques such as MRI enable large-scale, non-destructive imaging of entire tissues but may suffer from extensive acquisition times and relatively poor contrast resolution [31]. DVC offers three-dimensional, volumetric strain and displacement measurements throughout a tissue, enabling assessment of internal deformation under load [26]. While DVC provides valuable insights into internal tissue mechanics and heterogeneity, it typically requires high-resolution CT or X-ray imaging and can be computationally intensive.

Alternatively, strain based optical techniques such as digital image correlation (DIC), show great potential as a relatively inexpensive and scalable imaging

approach for assessing the load dependent fibrous network in collagenous tissues [32,33]. It has been employed in planar biaxial testing systems to measure mechanical properties of murine skin and determining the anisotropic material properties of knee ligaments [2,34]. Thus, DIC techniques are well suited towards examining the relationship between local structural properties and function of collagen structures in soft collagenous tissues.

Despite significant advances in high-resolution and microstructural imaging, there remains a need for an accessible approach that enables rapid quantification of collagen architecture in soft tissues across large fields of view (on the order of centimeters). Collagen fibres are the principal tension-resisting elements in connective tissues, and local strain can therefore serve as a quantitative marker of fibre alignment and recruitment under physiological loading. Although the bulk mechanical properties of soft tissues have been widely characterised using uniaxial tensile or compression tests, these methods do no capture the spatially varying, local structure-function relationships that govern tissue behaviour in vivo. Recent studies coupling biaxial mechanical testing with microstructural imaging have revealed strong regional dependencies between collagen orientation, dispersion, and mechanical response in tissues such as arteries, valve leaflets, and skin [2,17,18,29,30,35–37]. However, such systems often require specialised microscopy setups and are limited to small fields of view. This gap underscores the need for a method that combines multi-axial loading, homogeneous stress fields, and DIC to provide centimetre-scale, spatially resolved strain mapping that directly links local structure to mechanical function. Such an approach can also complement microstructural imaging techniques by linking tissue-scale mechanics with fibre-level organisation.

The characterisation of tissues with complex collagen structures, such as the hip capsule, provides a good platform for developing and testing the proposed opto-mechanical characterisation methodology. The hip capsule is generally described as consisting of three primary fibrous ligaments (iliofemoral, ischiofemoral, and pubofemoral) each providing functional roles to stabilise the hip joint [38–40]. While their structural anatomy and properties have been documented, there are mismatches in anatomical descriptions [41–43], quantitative data vary by up to an order of magnitude [44–47], and the deeper fibrous network has not been extensively studied [47]. These deeper circumferential collagen fibres commonly referred to as the zona orbicularis are reported to form a stabilising collar for the hip [48,49]. A comprehensive description of the hip capsule's fibrous network and mechanical properties is essential to advance surgical interventions [50–53]. Therefore, the aim of this study was to develop and demonstrate a biaxial testing approach that combines multi-axial loading with optical strain measurement to quantify the fibrous anatomy and spatial mechanical properties of complete tissues within a single biaxial test. Rather than introducing new imaging hardware, the contribution of this work lies in leveraging commercially available biaxial actuators and 2D DIC within a unified protocol designed to reveal load-dependent collagen organisation at the centimetre scale. The method was experimentally validated using an artificial anisotropic material and ex vivo skin, and was then applied to the human hip joint capsule.

## 2. Materials and methods

In this study, we quantitatively determine the fibrous anatomy and mechanical properties of the hip joint capsule complex using a characterisation method that captures the strain field while imposing a homogeneous biaxial stress distribution across the specimen.

### 2.1. Generation and control of a homogeneous biaxial stress distribution field

The homogeneous biaxial stress distribution field was generated using a custom-built multi-axial device featuring 22 pneumatic actuators (Indent, SMC Pneumatic, Tokyo, Japan) that can be operated in force-control (see S1 Appendix. for full parts list and assembly). Each actuator has a stroke length of 50 mm and a max operating load of 45N. This load range is appropriate for the present study because the imposed forces remain within the linear region of hip capsule behaviour. Published force–displacement data for the iliofemoral ligament, including those reported by Hewitt et al., show that the transition toward high stiffness and eventual failure occurs well above the 40 N loads applied here [54]. As shown

in ., where representative curves from Hewitt et al. are plotted alongside the forces generated in this study, our measurements fall within the initial linear region of the capsule response. This confirms that the 45 N actuator capacity is sufficient for accurate anatomical and mechanical characterisation without approaching failure loading. The actuators are arranged in an 8 by 3 format on an optical breadboard, allowing deformations to be imposed in up to 169 directions when actuators are operated in opposing directions (Fig 1a). A set of low friction drive pulleys accompanies each actuator to transmit load via 0.5 mm monofilament lines to the attachment point on the specimen (Fig 1b). Actuator control and image acquisition from the camera are controlled using a control system consisting of a microcontroller (MEGA 2560, Arduino, Boston, USA), an electronic pressure transducer (P31P, Parker Hannifin, Ohio, USA), a 24-array relay board (OKY3016, Switch Electronics, United Kingdom), and a solenoid manifold (V100, SMC Pneumatic, Tokyo, Japan). The actuators have constant piston cross-sectional area, and hence the force applied is controlled by an electronic proportional regulator that allows the outlet pressure to all actuators to be regulated proportionally by an electronic control signal. During the unloading operation, an additional relay module activates an exhaust solenoid valve connected via a manifold to the outlet port of all pneumatic actuators. This operation exhausts residual gases from the actuators and unloads the tissue.

The experimental setup consists of a DIC system (GOM Correlate Professional, Braunschweig, Germany) calibrated to track full-field strain measurements. A camera (1080p resolution, 30 fps, 750D Canon, Tokyo, Japan) positioned normal to the tissue surface acquires images of the entire tissue during the loading operation through intact and deformed states. A single camera was used for two dimensional DIC, meaning that only in plane strain components were measured. This setup is appropriate for the present study because the hip capsule specimens were mounted fully flat on a rigid frame and loaded strictly within their plane. The applied biaxial forces act tangentially along the specimen boundaries, which minimises out-of-plane deformation and ensures that the measured strain field accurately reflects the in-plane mechanical response of the tissue.

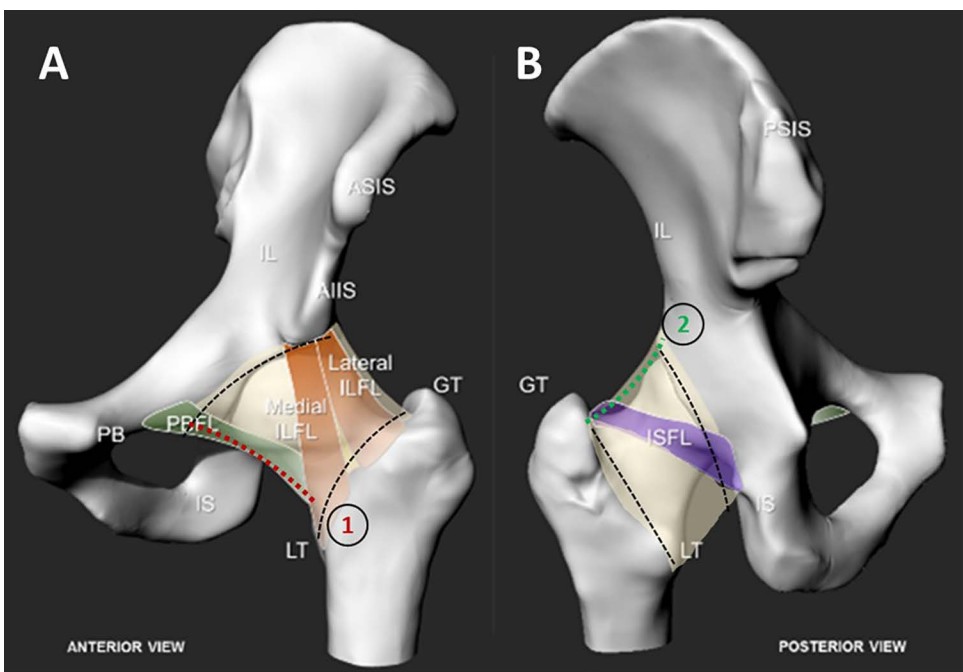

**Fig 1. Experimental setup: (a)** Custom-built opto-mechanical characterisation device, **(b)** Excised flat capsule specimen attached to 22 pneumatic actuators using the double hook gripping method and monofilament suture lines. Scale bars indicate a length of 25 mm.

## 2.2. Specimen preparation

Ethical approval for the study was provided by our institutions' ethics committee (Imperial College Healthcare Tissue Bank, United Kingdom) under study ID 19103. Nine hip joint capsules (n = 9, m/f = 6/3, age = 45 ± 9 years) were obtained from fresh frozen cadaveric specimens. Both male and female specimens were pooled for analysis, as previous work has reported no significant sex-related differences in the tensile properties of capsular ligaments within this age range [47]. Therefore, sex was not treated as a separate variable in this study. The anonymised specimens were used for experimentation between February 15th to 28th, 2023. Prior to excising the capsule, anatomical landmarks at femur and acetabulum attachment points were identified. A complete capsulectomy was performed by incising between one of these two areas: longitudinally between the Anterior Inferior Iliac Spine (AIIS) and Greater Trochanter (GT) or Lesser Trochanter (LT) and Transverse Acetabular Ligament (TAL), and then circumferentially to detach the capsule around the acetabular and femoral insertions to form a flat specimen (Fig 2). Two longitudinal cut locations were used so that anatomical information missing near one cut region, where strain data cannot be captured, could be recovered from the alternate region.

Capsules were prepared for DIC with a white-on-black speckle pattern: charcoal powder was used to give a uniform black background on which white titanium dioxide particles, corresponding to an ideal speckle size of 3–5 pixels, were dispersed to provide optimal contrast [55]. To ensure that the charcoal and titanium dioxide particles did not alter the natural mechanical behaviour of the capsule, we verified that these materials are inert, non-reactive and applied in quantities that do not measurably stiffen, hydrate or otherwise modify the capsule surface. The tissue was fully hydrated prior to speckle application, and no further hydration was applied during testing. The total acquisition time was under 300 s, which limited any risk of particle detachment or changes to the tissue surface. We also confirmed that the speckle pattern did not influence the mechanical response by comparing load–displacement curves obtained with two-point tracking, without a speckle pattern, against curves acquired with DIC using the powder-based pattern and found no detectable differences. The stability of the speckle pattern was further supported by the flat mounting configuration of the capsule, which minimised out-of-plane motion and prevented local loss of particle adhesion or correlation during deformation.

## 2.3. Homogeneous biaxial stress distribution test to quantitatively measure ligament anatomy

Specimens were mounted onto the device platform using 20 mm wide double fishhooks with uniform spacing along all four edges (eight actuators attached on femur/acetabulum edges and three actuators on the incised edges) and aligned to the testing axis to ensure homogeneous stress distribution during tissue characterisation.

A preload of 2.5 N was applied at corner actuators to remove laxity across the specimen. Based on previous work, a preconditioning regime was included to ensure a repeatable and consistent loading history between the hip capsule specimens [56]. A preconditioning regime was performed by subjecting all attachment points on the specimen to ten loading-unloading cycles up to a maximum load of 20 N at a loading rate of 1 N/s. At this loading rate, the resulting tissue deformation corresponded to an approximate strain rate of 0.01 s$^{-1}$. This value lies within a quasi-static, non-physiological regime and is comparable to the low strain rates used in previous characterisation studies of the hip capsule, including those reported by Hewitt et al. [54] and Pieroh et al. [47]. Using a quasi-static strain rate minimises viscoelastic rate effects and enables direct comparison with published capsule mechanics.

Following the preconditioning protocol and a preload stage, all tissue attachment points were loaded up to 40 N at 1 N/s to generate a homogeneous biaxial stress distribution. A total of 1200 images were captured during continuous tissue deformation and processed using DIC with a facet size of 9 pixels and 50% overlap to generate a minimum of 3000 strain points at each time frame. A minimum pattern quality of 0.4 was employed throughout and each pixel corresponded to 0.1 mm in the images. This resulted in an effective in-plane spatial strain resolution of approximately 0.45 mm. To quantify the systematic and random error of the DIC system, a zero-strain analysis was performed by acquiring images of a stationary capsule specimen under identical optical, lighting, and correlation settings. The resulting strain field showed mean values

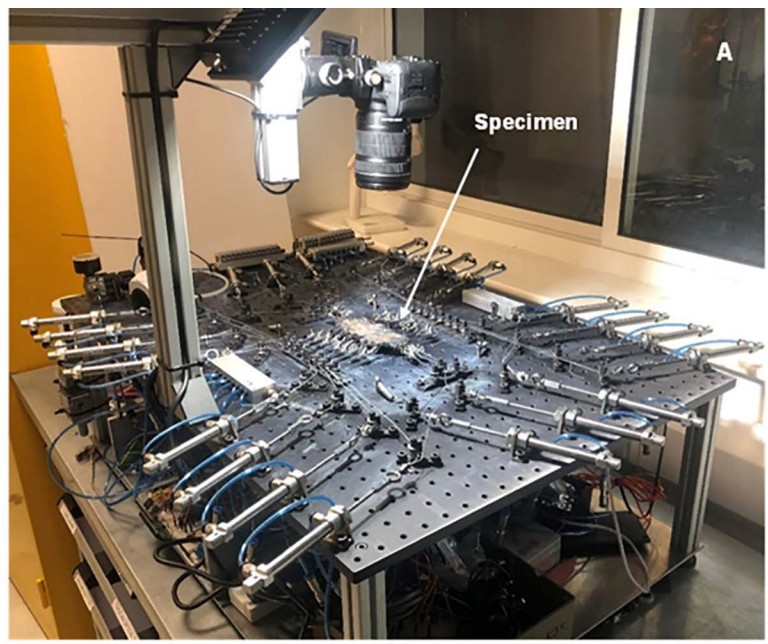

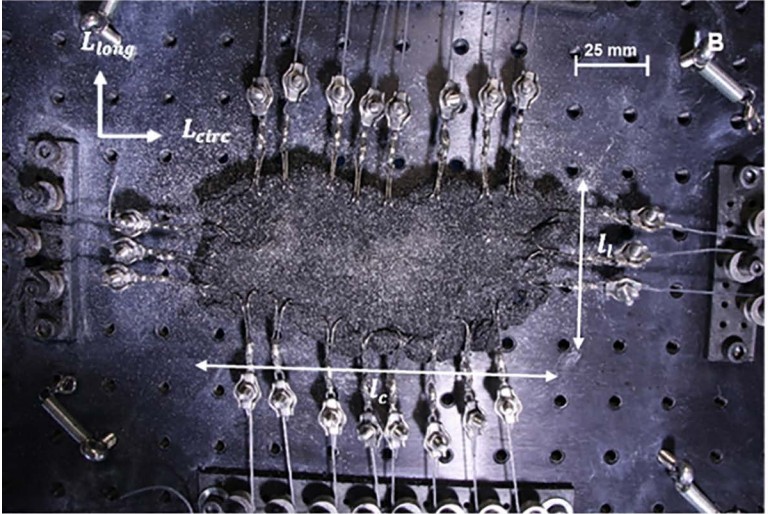

**Fig 2. (a) Anterior and (b) posterior views of the hip joint capsule complex.** The commonly identified ligaments are highlighted. Black dashed line denotes circumferential incision lines on acetabular and femoral insertions. Red and green lines denote two incisions to detach the capsule longitudinally; between greater trochanter and the anterior superior iliac spine (green); between lesser trochanter (LT) and pubofemoral ligament (red).

close to zero (approximately 0.02%) with random noise on the order of 0.10%, consistent with expected precision for the selected facet and overlap sizes. This confirmed that the DIC configuration did not introduce measurable artificial strain during acquisition.

The major strain field (%) defined as the strain in the direction of the maximum strain was used to quantitatively distinguish the load dependent fibrous anatomy in the capsule. All strain measurements reported in this study represent 2D in-plane strain fields obtained from a single-camera DIC system. Full 3D strain measurement would require a stereoscopic camera configuration, which was beyond the scope of the present work.

## 2.4. Hip capsular anatomy identification

The dominant fibrous anatomy amongst hip capsules tested was determined using a custom autocorrelation script implemented in MATLAB (R2018b; MathWorks, USA) (see S2 Appendix for sample code). The optimum alignment between the strain field matrices was detected by applying a matrix operation that calculates the mean square intensity of all specimen strain field matrices added together. This function is iteratively performed along the longitudinal $L_{long}$ and circumferential $L_{circ}$ axis (Fig 3).

Following the addition of all strain matrices, the alignment with the highest mean square intensity provided an output accentuating the regions where fibrous structures were present or absent. The number of structures and their location were predicted by analysing the deformation vector field. The local orientation and isotropic properties of the fibrous structures were derived from a 2 x 2 symmetric structure tensor generated for each pixel using ImageJ software and the OrientationJ plugin (Biomedical Imaging Group, EPFL, Switzerland). The structure tensor for each pixel was defined as a 2 x 2 symmetric positive matrix. The tensors were evaluated for the deformation field by computing the continuous spatial derivatives using a cubic B-spline interpolation method, resulting in an output matrix containing the orientation of all structural tensors. The local orientation ($\theta$) corresponded to the direction of the largest eigenvector of the tensor. A scanning window representing ligament dimensions was used to calculate the distribution of the orientation values across the field. The scanning window calculates the distribution of the orientation of structural tensors appearing in the region by iterating across rows and columns of the matrix at 0°. This operation was repeated at 1° increments up to 90° rotation by applying a 2D rotation transformation to the structural tensor matrix. The regions with highest orientation frequency peaks represented highly oriented structures and were identified as fibrous structures. The predicted angle ($\theta$) of each structure was calculated as the median of the distribution and reported relative to longitudinal ($L_{long}$) axis of the capsule. The detected structures were matched with anatomical definitions of ligaments previously reported [43,44].

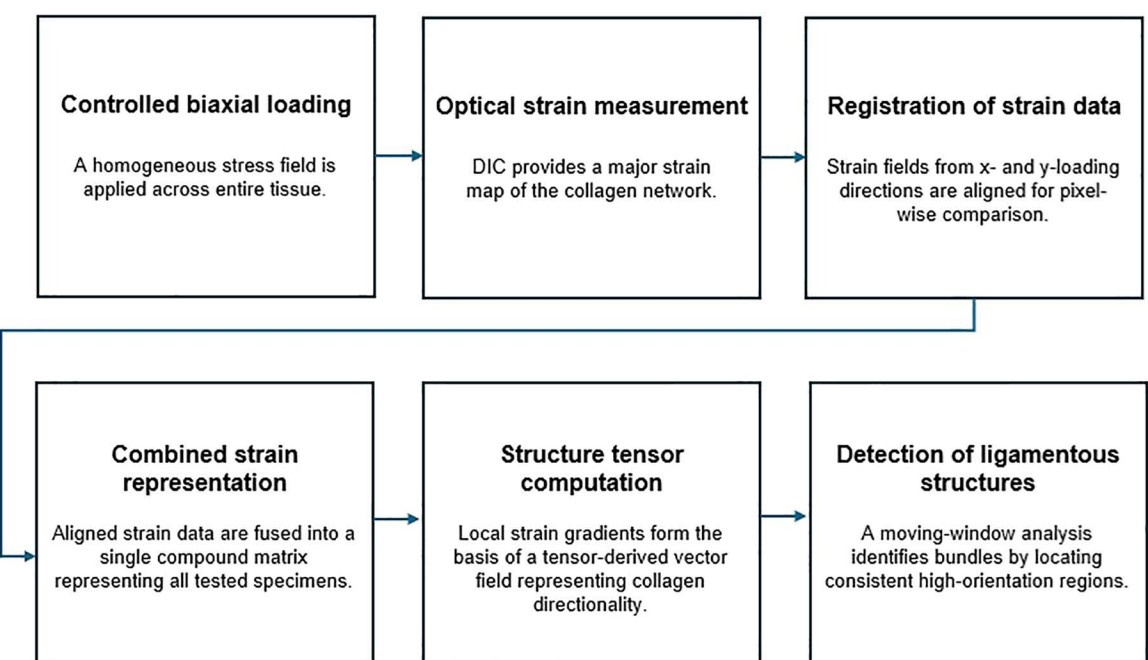

**Fig 3. A process flow of fibrous structure identification following opto-mechanical characterisation of the tissue.**

## 2.5. Tangent modulus and cross-sectional area of capsular ligaments

The mechanical properties of the structures identified using the strain-based method described in Section 2.4 were extracted in a single test by application of a homogeneous biaxial stress distribution. The ligaments were tested to determine their tensile behaviour at 1 N/s loading rate up to a maximum of 40 N in the testing direction. The widths of each detected ligament were measured as seen on surface strain measurements and thicknesses measured in-situ manually using an electronic thickness gauge in the preloaded state (Mitutoyo 547, 0.01 mm resolution). The mean ligament width and thickness were calculated by taking measurements at a distance of 25%, 50% and 75% along the ligament length. These dimensions were not used during anatomical characterisation but were required later to compute ligament-specific stress and tangent modulus, where true cross-sectional area is needed. The mean (± SD) cross-sectional areas of each ligament calculated for stress calculation purposes are reported in Table 1. The behaviour of the deformation for each fibrous structure was described using a widely adopted method [57]. The stretch ratios ($\lambda$) were calculated as follows:

$$\lambda_{iz} = \frac{l_{iz}}{L_z}$$

(1)

Where $\lambda_{iz}$ represents the local stretch (for every strain cell) of the ligament along its axis of displacement, $l_{iz}$ is the local length of the deformed region and $L_z$ is the local initial length of that region. The overall stretch for each ligament was determined by calculating the mean of the local stretches within which the stress is relatively homogeneous (S3 Fig.).

The effect of shear components on in-plane ligament tension was investigated by comparing ligament strain during uniaxial and biaxial loading scenarios and verified for one capsule. The ligaments were uniaxially tested by loading select actuators to 40 N along the detected ligament axis. The strain (%) generated along the ligament long axis was compared for the two loading scenarios. The difference in strain (%) between the uniaxial and biaxial loading scenarios was less than 5%, indicating that shear components did not substantially influence the strain measured along each ligament. This comparison was performed only to confirm that shear effects did not confound strain extraction along the ligament axis. Biaxial loading remains essential for the proposed method because the capsule contains multiple fibre families with different orientations, and only a homogeneous biaxial stress field can uniformly load the entire fibrous network and reveal its spatial anatomy. Given the assumption of incompressibility ($\lambda_x\lambda_y\lambda_z$ = 1) and negligible shear components, the Cauchy stress for each detected ligament was calculated as follows:

$$\sigma = \frac{f_z\lambda_z}{tW}$$

(2)

Where $f_z$ and $t$ are the measured forces and thickness, respectively, $\lambda_z$ is the measured stretch from DIC along the ligament axis, and $W$ is the mean ligament width from surface strain measurements perpendicular to the direction of loading.

**Table 1. The mean cross-sectional area (± SD) of hip capsular ligaments.**

| Ligament | Mean ligament cross-sectional area (± SD) (mm²) |
|---|---|
| Lateral Iliofemoral | 66.8±4.61 |
| Medial Iliofemoral | 43.6±10.5 |
| Pubofemoral | 31.7±4.8 |
| Superior Ischiofemoral | 23.2±5.3 |
| Inferior Ischiofemoral | 15.3±6.2 |
| Zona Orbicularis | 15.6±3.3 |

A tangent modulus was determined by calculating the slope based on the last 10% of stress values over the high stiffness domain of the Cauchy stress-stretch curves for each identified ligament. All linear fits of the high stiffness region had $R^2$ values >0.98 and, thus, were representative of the experimental data.

## 2.6. Sensitivity analyses

Four sensitivity analyses were conducted to verify the performance and robustness of the opto-mechanical characterisation system (see S3 Appendix for details).

1) Verification of homogeneous biaxial stress generation

The presence of a biaxial stress distribution was verified using two elastomeric parts, one isotropic and one anisotropic material, each with dimensions of 140 mm x 50 mm x 3 mm and printed with a Young's Modulus of 1.5 GPa. The anisotropic part contained thin channels forming the letter "K". The parts were speckled using an airbrush technique previously reported by Palanca et al. [58] mounted onto the device platform using the same attachment configuration and testing protocol described in Section 2.3. This analysis confirmed that the actuator array can generate a uniform biaxial stress field across the specimen surface, ensuring that strain measurements represent material response rather than non-uniform loading.

2) Assessment of boundary and gripping effects

The effect of this fishhook gripping method was evaluated by quantifying the Saint-Venant boundary effects in a finite element model. Simulations of the stress distribution field generated by the multi-axial device setup were performed in ABAQUS/Standard 6.13 (Dassault Systèmes Simulia Corp., Rhode Island, USA). The resulting maximum principal strain fields were compared with the DIC strain field measurements, and the Saint-Venant effects were quantified along the centreline of the specimen to identify any boundary influence extending toward the central region. This analysis determined whether the gripping approach or edge constraints altered stress homogeneity, confirming that local strain variations within biological specimens were intrinsic rather than artefacts of boundary conditions.

3) Demonstration of applicability to other biological tissues

The characterisation method was applied on a skin sample harvested from a cadaveric shoulder specimen to demonstrate its applicability to other biological tissues. A homogeneous biaxial stress distribution was applied across the sample with same tissue attachment configuration and loading conditions used for the capsule tests. The major strain field (%) was calculated and macro scale fibrous structures were analysed. This analysis demonstrated the versatility of the technique for different planar collagenous tissues such as skin, fascia, and membranes.

4) Evaluation of robustness through actuator removal testing

A leave-one-out technique was used to assess the robustness of the stress distribution field produced by the multi-axial device. A total of 22 tests were run, each time one actuator attachment was left detached from the tissue before running the test. The mean major strain (%) was calculated for each scenario, and the mean percentage error was used to evaluate the technique. This analysis showed that the stress distribution field and strain measurements remain stable even with minor deviations in actuator loading, confirming the system's robustness.

These analyses verified that the testing system produces homogeneous loading and is not affected by boundary conditions along the specimen centreline. They also confirmed that the method can be applied to different soft tissues and maintains stable performance even when actuation symmetry is altered.

## 2.7 Statistical analysis

Data are reported as a mean±standard deviation. Statistical analysis on tangent moduli differences between ligaments was conducted with Prism software (GraphPad Prism 8.2.1, US). The normality of the test group was assessed

using Shapiro-Wilk tests before further analysis. If normality was not satisfied, equivalent non-parametric tests (Friedman test with Dunn's multiple-comparison post hoc) were applied. Normally distributed data were analysed using a repeated-measures one-way ANOVA and post-hoc Tukey test, with significance set at $p < 0.05$.

## 3. Results

### 3.1. Hip capsular anatomy identification

The mean size of the capsule was measured to be $122 \pm 12$ mm and $38 \pm 8$ mm in the projected longitudinal $L_{long}$ and circumferential lengths $L_{circ}$, respectively. Under the imposed loading condition, the mean applied stress across the capsule surface was approximately 1.2 MPa. The (%) major strain across the fibrous anatomy was $7\% \pm 2\%$. A prominent non-fibrous region was identified as the iliopectineal bursa with a major strain of $85\% \pm 17\%$. Further non-fibrous regions localised between the longitudinal and circumferential fibrous regions were identified with a major strain of $42\% \pm 9\%$.

A total of seven fibrous structures were revealed from the compound strain map (n = 9) (Fig 4a). The structural tensors unique to each structure were presented in probability density histograms (Fig 5) and used to verify the ligament locations and their orientation. The colour gradient represented the presence of fibrous regions across the stacked capsule; red representing regions with fibrous structures detected across all capsules and black representing regions detected with no fibrous structures in any capsule. The green and yellow regions depicted regions where fibrous structures were present in some but not all capsules. In the anterior capsule, the lateral iliofemoral ligament (structure 5) and pubofemoral ligament (structure 3) had fibrous structures predominantly aligned close to the longitudinal $L_{long}$ axis (median $\theta_{fiber}$ = 27° and median $\theta_{fiber}$ = 18° respectively). The medial iliofemoral ligament (structure 6) had a median fibrous structure orientation of 43°. A 3D representation of the anterior and inferior hip capsular anatomy is provided in Fig 4b and 4c, respectively.

In the posterior capsule, the orientation of the fibrous structures for the inferior ischiofemoral ligament bands (structure 1 and 2) were predominantly aligned to the $L_{long}$ axis (median $\theta_{fiber}$ = −33°, median $\theta_{fiber}$ = −19°). The superior ischiofemoral ligament (structure 4) had a median fibrous structure orientation of 40°.

A circumferential fibrous region (structure 7), commonly referred to as the zona orbicularis (ZO), formed confluency between the longitudinal structures of the capsule. This circumferential structure shared multiple populations of fibrous bundles with the six detected longitudinal structures. The orientation of structural tensors within the scanning window (including those for fibrous structures) had a higher probability density that aligned close to the circumferential axis (median $\theta_{fiber}$ = 97°).

### 3.2. Tangent modulus and cross-sectional area of capsular ligaments

All datasets passed normality testing (Shapiro–Wilk, $p > 0.05$), confirming that the use of parametric analyses was appropriate. The mean Cauchy stress-stretch curves for the capsular ligaments are shown in Fig 6. The mechanical behaviour of the ligaments was nonlinear with a soft toe region transitioning into a stiffer higher stress region. The measured mean cross-sectional areas of the six identified ligaments are reported in Table 1. The hip capsule was found to be anisotropic (Fig 7). The tangent modulus of the zona orbicularis was 65% larger than the medial iliofemoral ligament ($p \le 0.01$), and 66% larger than the pubofemoral ligament ($p \le 0.05$). The lateral iliofemoral ligament was found to have 44% larger modulus than the pubofemoral ligament ($p \le 0.05$).

### 3.3 Sensitivity analyses

#### 1) Verification of homogeneous biaxial stress generation

The mean major strain across the primary region of the isotropic and anisotropic parts was $3.15\% \pm 0.4\%$ and $3.25\% \pm 0.6\%$, respectively (see S2 Fig.). In the channel regions of the anisotropic part, the major strain increased to

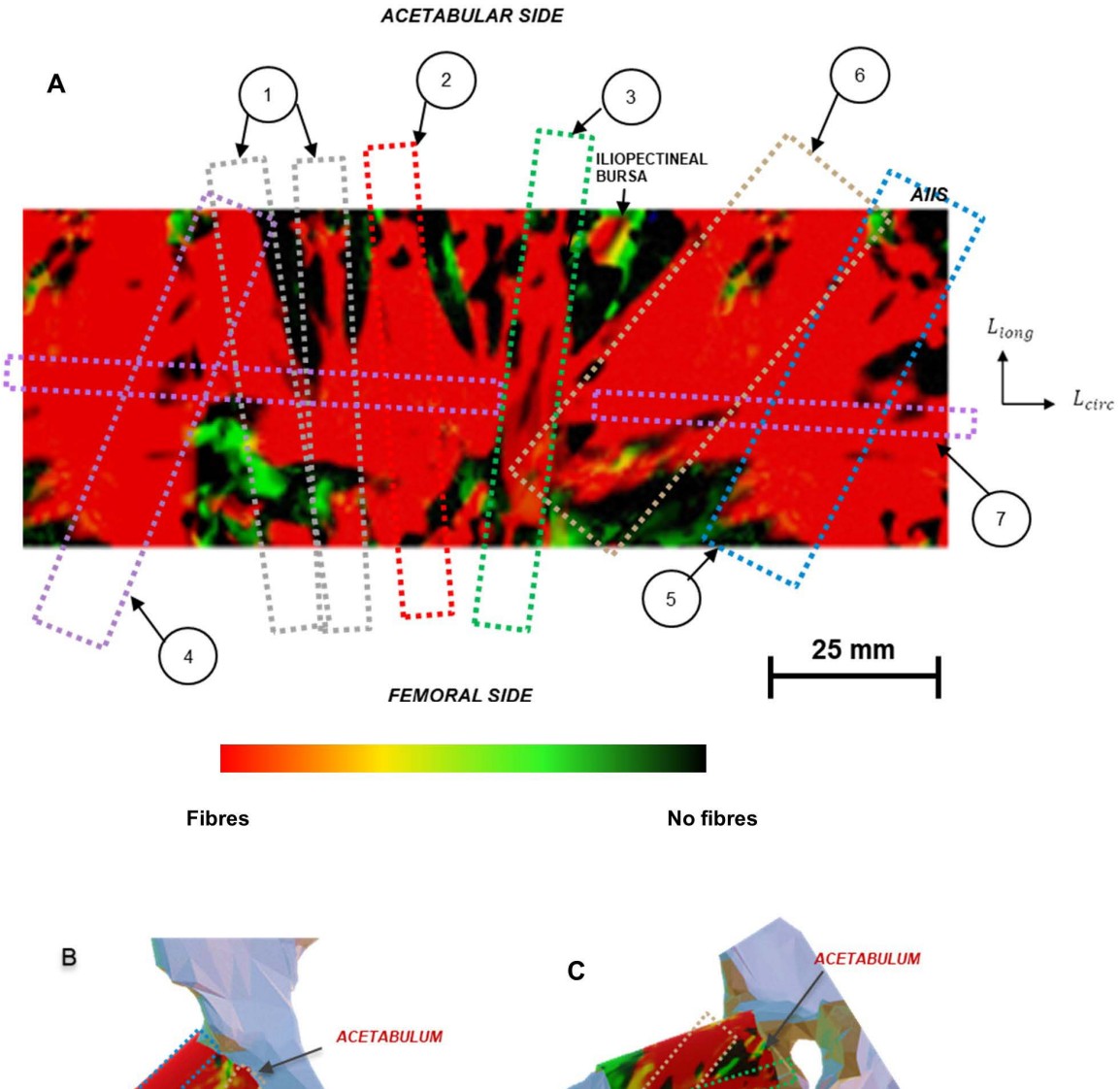

**Fig 4. (a) Image of detected fibrous structure network (represented by red contour) of human hip joint capsule revealing seven ligamentous structures (Image is a stack of n = 9 measured hip capsules), (b) Anterior view of fibrous network reconstructed onto right hip joint anatomy, (c) Inferior view of fibrous network reconstructed onto right hip joint anatomy.** AIIS, Anterior Inferior Iliac Spine; GT, Greater Trochanter; LT, Lesser Trochanter. The colour scale represents the presence of fibrous structures across the stacked capsule, red representing regions with fibrous structures detected across all capsules and black representing regions detected with no fibrous structures in any capsule. Numbering corresponds to the probability density histograms shown in Fig 5, which were used to identify the ligaments and their orientation. Scale bars indicate a length of 25 mm.

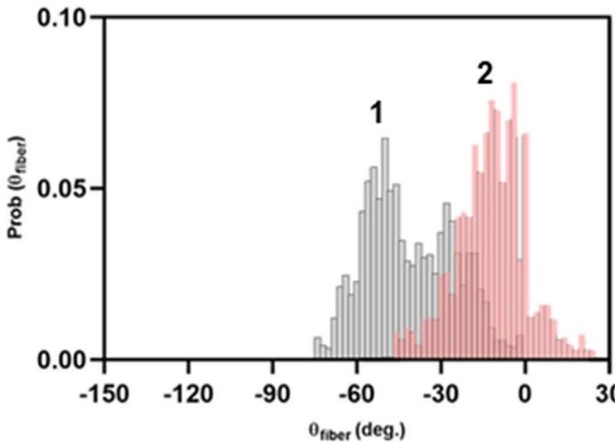

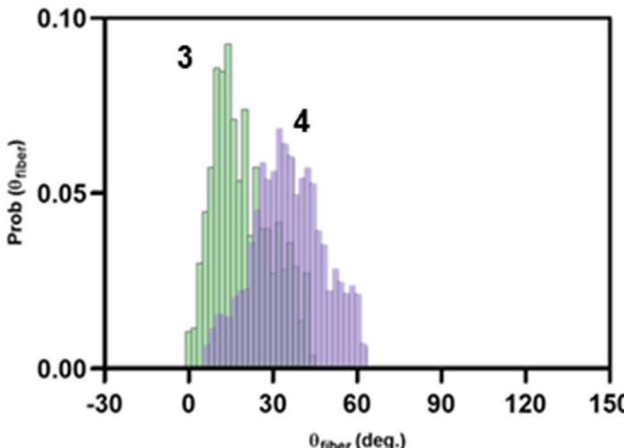

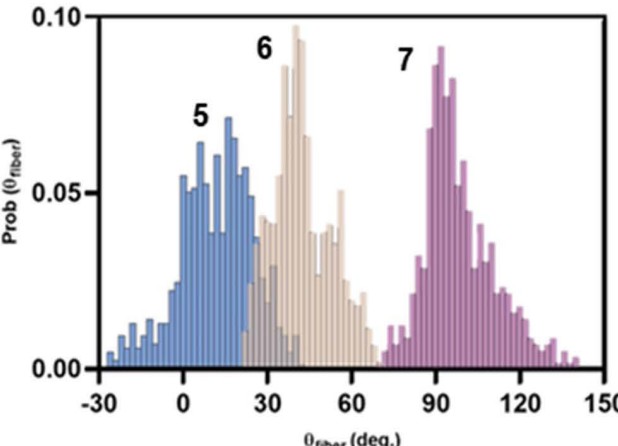

**Fig 5. Ligamentous structures detected from probability density histograms of fibre bundle orientation, relative to $L_{long}$ axis.** Numbering corresponds to structures denoted in Fig 4.

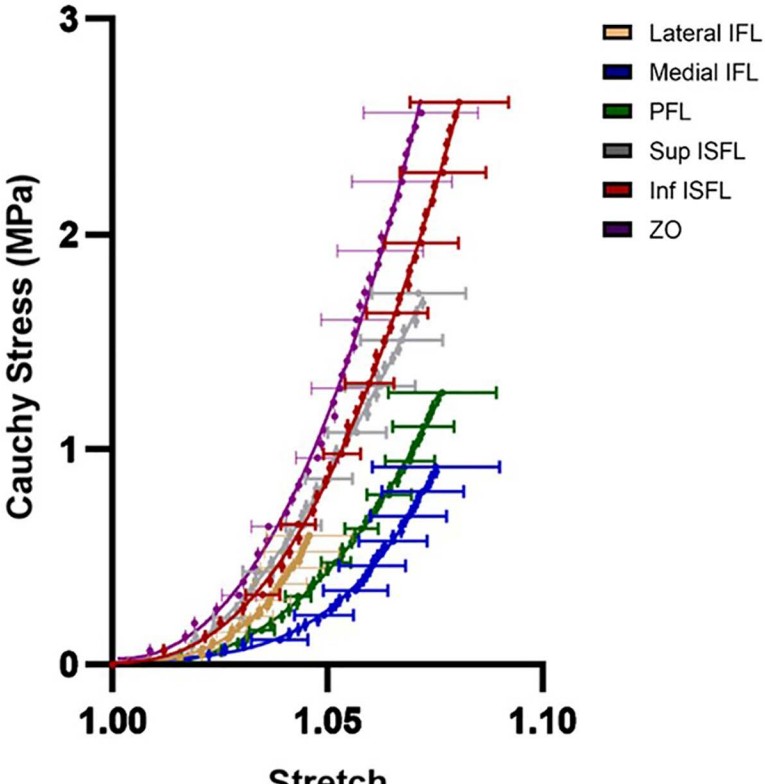

**Fig 6. The mean ligament stress versus stretch for identified ligaments in the hip capsule.** (n = 9 per ligament, the error bars are standard deviation). IFL, iliofemoral ligament; PFL, pubofemoral ligament; ISFL, ischiofemoral ligament; ZO, zona orbicularis.

6.7% ± 0.2%, approximately twice that of the surrounding regions due to a 50% reduction in cross-sectional area. The "K" shaped channel region was visually verified from the strain fields, confirming that the device generated a homogeneous biaxial stress field that accurately reflected material behaviour.

### 2) Assessment of boundary and gripping effects

Finite element analysis showed that stress decay along the specimen diminished at 8 mm away from the grip edges. The central region, covering approximately 88% of the width between the grips in the circumferential axis and 65% in the longitudinal axis produced uniform stress distribution free of boundary effects (see S3 Fig.). These findings confirmed that the fishhook gripping method did not distort the internal stress field and that Saint-Venant effects were confined to the specimen edges.

### 3) Demonstration of applicability to other biological tissues

Application of the opto-mechanical method to skin demonstrated its capability to capture skin's anisotropic nature, revealing the fibrous network and the well-recognised Langer's Lines (see S4 Fig.). This confirmed the method's suitability for characterising other planar collagenous tissues beyond the hip capsule.

### 4) Assessment of system robustness through actuator removal

The leave-one-out analysis demonstrated that the characterisation method could predict the major surface strain with a mean squared error of 4% (see S5 Fig.). The stress distribution field and strain measurements remained consistent even when one actuator was detached, confirming the robustness and stability of the system under minor changes in actuation.

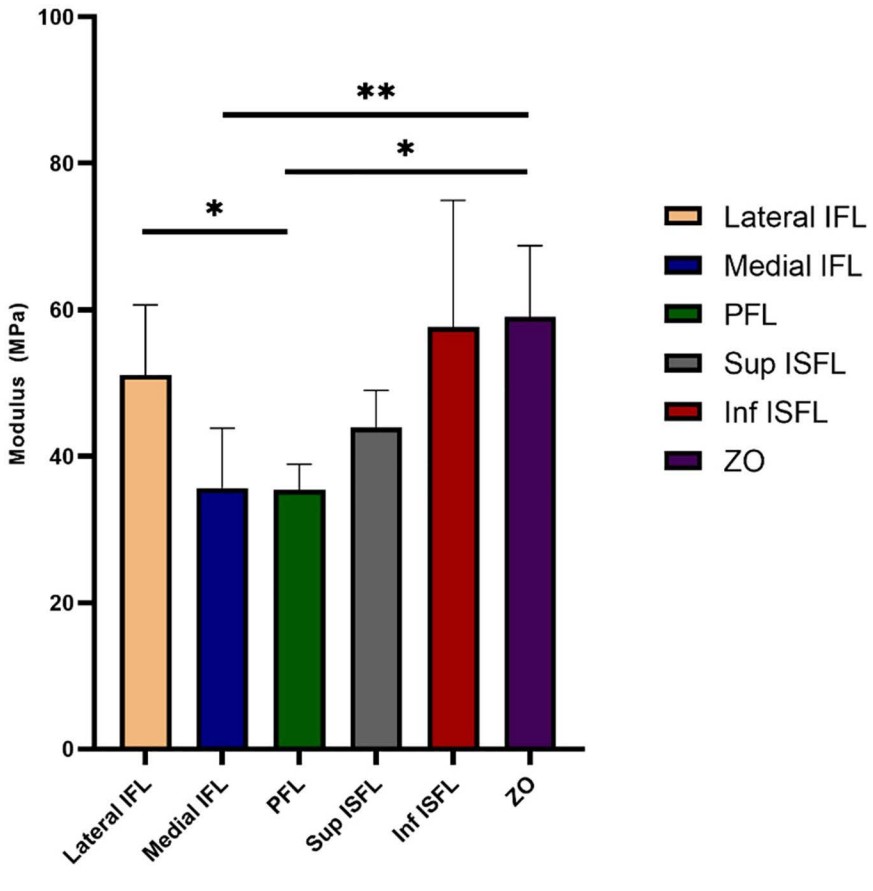

**Fig 7. The mean tangent modulus with standard deviation for identified ligaments in the hip capsule extracted in a single test by application of a homogeneous biaxial stress distribution.** (n = 9 per ligament). Statistical significance is indicated as follows: * p ≤ 0.05, ** p ≤ 0.01. IFL, iliofemoral ligament; PFL, pubofemoral ligament; ISFL, ischiofemoral ligament; ZO, zona orbicularis.

## 4. Discussion

The most important finding of the study was that our proposed characterisation method was able to quantitatively describe the structural organisation of highly anisotropic, planar soft collagenous tissues at the tissue level. We applied this method to the hip joint capsule and simultaneously characterised its fibrous network and material properties. This opto-mechanical analysis method could have broad applications across ligaments, organs, arteries, and skin. The hip joint capsule was found to be mechanically heterogeneous, with seven distinct fibrous structures forming a wire-mesh-like network. Using a single biaxial test under a verified homogeneous stress field, we distinguished macro-scale fibrous regions from non-collagenous regions such as the iliopectineal bursa, using major tissue strain as a quantitative measure. Under uniform biaxial loading, regions of lower strain corresponded to the load-bearing collagen network, consistent with the principle that collagen fibres are the main tension-resisting elements in connective tissues. Local orientations ($\theta_{fiber}$) revealed the principal directions of these fibrous regions and their lines of action. The tangent modulus of the detected fibrous structures ranged between 35 MPa to 60 MPa, demonstrating the mechanically heterogeneity of the capsule.

The novelty of this study lies in the integration of several key methodological elements within a single, accessible framework. The approach combines biaxial mechanical loading, homogeneous stress verification, DIC, and

centimetre-scale imaging to enable simultaneous characterisation of structure and function without disrupting native tissue architecture. Biaxial loading provides a controlled mechanical environment that captures multi-directional fibre orientations and their spatial distribution across the sample. Establishing a homogeneous stress field ensures that regional strain differences reflect intrinsic material behaviour rather than boundary artefacts. The use of DIC provides local spatial resolution, enabling detailed mapping of strain fields and quantitative inference of collagen fibre orientation and recruitment. By preserving the anatomical continuity of the specimen and avoiding sectioning or fixation, the method maintains structural relevance and links regional fibre organisation directly to mechanical function. The large field of view bridges the gap between microscopic imaging and bulk mechanical testing, while the simultaneous derivation of local tangent moduli provides direct quantification of mechanical properties within the same experiment. Each of these elements has been applied independently in earlier studies, but their integration here creates a unified experimental system capable of mapping the local structure–function relationship across entire tissues. This combination represents a significant advance over conventional testing approaches.

Optical–mechanical systems have significantly advanced understanding of collagen architecture under load by coupling imaging modalities with biaxial or uniaxial testing. Early work by Sacks and colleagues [26,28] combined biaxial mechanical testing with SALS to quantify collagen fibre orientation and realignment in soft tissues. Subsequent systems, including those described by Yang et al. [20], Blair et al. [59], Jett et al. [18], and Laurence et al. [29], expanded this principle using advanced imaging techniques such as pSFDI and PLI to characterise collagen architecture during deformation. More recent studies have employed SHG microscopy to link microstructure and function, including layer-specific investigations in the bovine trachea [30], mitral valve leaflets [35], and regional mechanical–structural characterisation of mouse skin [2]. Related biaxial investigations have also examined the creep behaviour of cardinal ligaments [36] and structural differences in the iliac arteries [37]. These techniques achieve high angular precision but rely on laser optics, polarisation control, and thin, transparent samples, which restrict their field of view to sub-millimetre or millimetre-scale regions. In contrast, the present DIC-based framework infers structural orientation from surface strain fields across centimetre-scale areas under a homogeneous biaxial stress, eliminating the need for specialised optical hardware. This enables quantitative assessment of fibre orientation and regional stiffness while revealing the functional anatomy of the tissue from its strain-derived fibrous network. The method therefore complements SALS, pSFDI, and PLI systems by extending their insights to the tissue scale within a single, accessible experiment.

The strain-based vector field used in the structure detection algorithm revealed the spatial organisation of the hip capsule's fibrous collagen network. Regional strain variations and fibre directions corresponded closely with previous qualitative fibrous descriptions for the iliofemoral, pubofemoral and ischiofemoral ligament bands [43,60,61]. The identification of a circumferential fibrous region orientated near θ = 0°, corresponding to the zona orbicularis, aligns with previous anatomical descriptions of fibres encircling the femoral neck from the anterior to the posterior aspect [43] and supports its proposed role as a stabilising collar that resists edge loading [49]. The calculated tangent moduli, ranging from 35 MPa to 60 MPa, align with but narrow the wide range previously reported from uniaxial tests of 10 MPa to 100 MPa [44,54]. This difference is likely due to the ability of biaxial loading to avoid malalignment of deeper collagen fibres. More recent uniaxial data from Pieroh et al. and Schleifenbaum et al. [46,47] reported lower moduli below 30 MPa, reinforcing the importance of multi-axial characterisation before determining material properties. Regional differences in modulus, such as higher stiffness in the zona orbicularis region and lateral iliofemoral ligament, may reflect local variations in fibre density and alignment. These findings demonstrate the capability of the method to distinguish local material properties within complex tissue structures.

Traditional macro-scale testing methods often overlook the heterogeneity and anisotropy of the hip capsule. Earlier studies required physical sectioning or chemical fixation of ligament bundles for uniaxial tensile testing or histological analysis, which can alter fibre alignment and introduce error [44,46,47,54]. By testing the capsule as a single continuous structure, the current method preserved tissue integrity while capturing spatially resolved mechanical behaviour. It removes the

need for multiple complementary tests and complicated data processing typically associated with planar tissue characterisation [18,22,62]. Although high-resolution optical techniques such as SHG and confocal microscopy, and PLI remain valuable for microstructural insight, their small field-of-view limits whole-tissue assessment. The current method provides centimetre-scale, single-test characterisation that can be integrated with microstructural imaging in a complementary multiscale framework.

Several limitations should be acknowledged. Although the setup enables simultaneous measurement of heterogeneity and mechanical properties, higher-resolution imaging would be required to capture fibril-level micromechanics. The automated algorithm is largely resolution independent, provided that DIC speckles are three to five pixels in diameter, but repeating tests at multiple load levels could improve strain sensitivity. The gripping method allows lateral expansion, although discrete groups near the attachment points may carry disproportionate loads. However, the leave-one-out sensitivity analysis showed minimal error, with a mean-squared error of around 4%. Biaxial stress methods are most effective for thin tissues such as unwrapped capsule, stomach, fascia, skin. For thicker samples, out-of-plane fibre engagement may cause minor deviations in stiffness estimation. Future refinements, such as testing at physiological temperature or using adaptive stress-homogeneity algorithms for irregular geometries, could further enhance performance. Applying this approach to tissues such as skin, stomach, and other joint capsules will help establish population-level maps of local structure-function relationships.

This study presents a new framework that combines biaxial mechanical testing, homogeneous stress verification, DIC-based local strain mapping, and centimetre-scale imaging to quantify the relationship between structure and function in soft collagenous tissues. Application to the hip capsule revealed the complexity and regional organisation of its fibrous network, providing new insights into regional mechanical roles. The method complements existing microstructural imaging techniques, extends biaxial-microstructure approaches to larger scales, and offers a scalable route for analysing healthy, diseased, or engineered tissues to guide improved treatments and design strategies.

## Supporting information

**S1 Appendix. Opto-mechanical characterisation device parts lists and assembly.**
(PDF)

**S2 Appendix. Example code showing post processing algorithm for detecting fibrous structures.**
(PDF)

**S3 Appendix. Sensitivity analyses to verify the performance and robustness of the opto-mechanical characterisation system.**
(PDF)

**S1 Fig. Representative force–displacement curves from this study compared with published iliofemoral ligament data [54].** The 'X' markers indicate the approximate failure region from Hewitt et al. All loads applied in this study lie within the initial linear region and remain well below reported failure levels.
(TIF)

**S2 Fig. Anisotropic parts deformed under homogeneous biaxial stress distribution: a) experimentally tested using the opto-mechanical characterisation device, b) Finite element analysis of equivalent part under same loading conditions.**
(TIF)

**S3 Fig. The stress decay along the centreline was quantified using the von Mises stress.**
(TIF)

**S4 Fig. The major strain (%) for skin specimen subject to a uniform biaxial stress distribution.**
(TIF)

**S5 Fig. The mean (%) major surface strain after removal of a selected specimen node.** Red line represents the mean (%) major surface strain with all specimen nodes attached.
(TIF)

## Author contributions

**Conceptualization:** Kabelan J Karunaseelan, Sarah K. Muirhead-Allwood, Richard J. van Arkel, Jonathan R.T. Jeffers.

**Data curation:** Kabelan J Karunaseelan.

**Formal analysis:** Kabelan J Karunaseelan.

**Funding acquisition:** Jonathan R.T. Jeffers.

**Investigation:** Kabelan J Karunaseelan.

**Methodology:** Kabelan J Karunaseelan, K.C. Geoffrey Ng, Richard J. van Arkel, Jonathan R.T. Jeffers.

**Supervision:** K.C. Geoffrey Ng, Sarah K. Muirhead-Allwood, Richard J. van Arkel, Jonathan R.T. Jeffers.

**Visualization:** Richard J. van Arkel, Jonathan R.T. Jeffers.

**Writing – original draft:** Kabelan J Karunaseelan.

**Writing – review & editing:** Kabelan J Karunaseelan, K.C. Geoffrey Ng, Sarah K. Muirhead-Allwood, Richard J. van Arkel, Jonathan R.T. Jeffers.

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
