## [Decision Letter · Decision Letter 0]

13 May 2025

Dear Dr. Karunaseelan,

Thank you for submitting your manuscript to PLOS ONE. After careful consideration, we feel that it has merit but does not fully meet PLOS ONE’s publication criteria as it currently stands. Therefore, we invite you to submit a revised version of the manuscript that addresses the points raised during the review process.

We look forward to receiving your revised manuscript.

Kind regards,

Alessandra Aldieri

Academic Editor

PLOS ONE

Journal Requirements:

2. Thank you for stating the following financial disclosure: [Funding for this research was provided by the National Institute for Health Research (NIHR300013)..

6. We are unable to open your Supporting Information file [Supplementary Material 1.docx]. Please kindly

Reviewers' comments:

Reviewer's Responses to Questions

**Comments to the Author**

1. Is the manuscript technically sound, and do the data support the conclusions?

Reviewer #1: Yes

Reviewer #2: Yes

2. Has the statistical analysis been performed appropriately and rigorously?

Reviewer #1: Yes

Reviewer #2: No

3. Have the authors made all data underlying the findings in their manuscript fully available?

Reviewer #1: Yes

Reviewer #2: No

4. Is the manuscript presented in an intelligible fashion and written in standard English?

Reviewer #1: Yes

Reviewer #2: Yes

Reviewer #1: In the paper: Multi-axial strain mapping to characterize structure and material properties of the human hip joint capsule. The Authors used a 2D DIC system and a biaxial testing machine to characterize the 2D local strain distribution of the hip joint capsule. The work is interesting but needs some improvements so I suggest Major Revision to the manuscript.

1) Abstract and Introduction in general: The Authors mention the innovative nature of their method but, in the Reviewer’s opinion, this take-home-message can be a bit speculative since, in the end, is a 2D DIC commercial system, with images acquired with a commercial photo camera, equipped on a bi-axial multi-actuator testing machine. The Reviewer suggests rephrasing the abstract/end of the introduction (or where appropriate) giving more evidence to the bi-axial machine instead of describing the setup as an innovative technique.

2)Introduction, lines 62-76: among the different imaging and X-ray/ strain-based techniques please add also the digital volume correlation approach and explain the pros-cons of this technique.

3) Materials and methods:

- lines 113-128: The Authors state that the load cells of the arms are 45N which is a quite low end-scale for characterising the mechanical behavior of ligaments up to failure. Please evaluate to add the typical force-strain curves of the hip joint capsule showing where your data falls into that curve. Please, if you acquire new data, be consistent with the strain-rate used for your tests, otherwise report literature data but explain possible differences with the strain-rate applied between the Authors’ and the literature data.

- please clearly report the strain-rate applied for your tests since, being ligaments viscoelastic, this parameter is fundamental to evaluate possible stiffening or softening of the material. Please also state if the strain-rate values used were physiological or not. Please add this data close to the 1 N/s value the Authors reported as speed.

- Lines 129-136: looking at the image of the system, just one camera was used so the local strain information is just 2D. Please clearly state for the Inexpert Reader and argument how you overcome the problem of possible tissue present in different planes during the acquisition.

- lines 151-155: about the speckle pattern: did the Authors test that the powder and the titanium dioxide did not compromise the natural tissue properties? This information is fundamental to validate our method. Moreover, being a speckle pattern based on particles and powder, how the Authors were sure to not lose locally the correlation during the hydration of the tissue or the possible detachment of the powder. Please clarify also in the manuscript.

- lines 170-175: To have an overall idea of the systematic and random error of the DIC investigation did the Authors performed a zero-strain analysis? Please clarify in the manuscript.

4) general remark: is important to clearly mention in the manuscript, for the Inexpert Reader, that all the local strains described in the text via this DIC system are only 2D. To obtain a 3D DIC investigation a minimum of 2 cameras are needed.

Reviewer #2: MAJOR COMMENTS

In the abstract, the authors at lines 33-34 say “We experimentally validate our method by applying it to the hip joint capsule”, while they say nothing about skin and artificial anisotropic material. Actually, from the main text, it seems that validation is against skin and artificial material, while hip joint capsule is the focus application. Therefore, this passage should be revised and briefly integrated.

The novelty of the proposed technique should be better demonstrated. For example:

i. a detailed comparison is expected vs references (18, 26-28) and (17), overall vs (26-28) and (17) that are cited in Introduction but not in Discussion;

ii. the study is based on the statement “the direct link between the local structural properties and tissue function need to be further investigated (33). This gap…”. However, a single, not recent (2015) paper (ref. 33) is not enough to support the whole rationale of this study. Check for a more recent bibliography, here and in general;

iii. definitely, more explicit support should be given in the manuscript to demonstrate the novelty of the technique and its impact. Which are the designed advantages respect to the state of the art? In a dedicated Discussion section, the key-elements – i.e. “biaxial”, “homogeneous stress”, “anatomy/structure”, “mechanical properties”, “local”, “large field of view” – could be declared, singularly highlighted in their importance and then the authors could demonstrate that their technique, combining those elements, represents an innovation and an advancement respect to the most recent solutions: e.g., refer and compare to:

• Microstructure and mechanics of the bovine trachea: Layer specific investigations through SHG imaging and biaxial testing (2022)

• Mechanical, structural, and morphological differences in the iliac arteries (2024)

• Biomechanics of mitral valve leaflets: Second harmonic generation microscopy, biaxial mechanical tests and tissue modelling (2022)

• The regional-dependent biaxial behavior of young and aged mouse skin: A detailed histomechanical characterization, residual strain analysis, and constitutive model (2019)

• Effects of repeated biaxial loads on the creep properties of cardinal ligaments (2017)

In “2.2 Specimen preparation”, the authors speak about nine harvested capsules, but in the rest of the text, it seems that only six are tested. Please, clarify this discrepancy, and comment about the m/f = 6/3 ratio in relation to the homogeneity of the tested group (i.e., hypothesis of no difference between males and females? Or only males tested?).

As for the general aim, the specific aims of paragraph 2.6 “Sensitivity analyses” should be more explicit in the text. That is, for each one of the analyses 1), 2), 3) and 4), to say why and why it is important to do (i.e., what their results implicate).

Related to 2.7 “Statistical analysis”, what if data were not normal? It should be taken into consideration; moreover, authors do not say that data resulted normal in 3 “Results”paragraph.

MINOR COMMENTS

Abstract

• Line 37, please briefly specify how strain measures anatomy

Introduction

• An introduction figure could be useful, a sort of graphical abstract

• Line 44, “play” instead of “plays”

• Line 45, check the position of the dot after “(1)” and in the rest of the manuscript

• Line 46, “comprises” instead of “comprise”

• Line 47, “is” instead of “are”

• Lines 54-61 underline the importance of the load-dependent nature of the fibrous collagen network; but this work does not really investigate this concept, i.e. a load-dependent arrangement. Please, clarify

• Line 63, what is “tissue-level”? Please, revise the statement

• Lines 74-76, some comments could be added in relation to contrast-enhanced microtomography (e.g., see https://doi.org/10.3390/biomimetics9080477)

• Line 86, “,” instead of “;”

• Line 90 ,“needs” instead of “need”

• Line 95, please specify what is “this new methodology”

• Line 96, “three” word better than “3” number

Materials and Methods

• Line 115, move “Fig.” before “S2”, renumber figures in order of citation (e.g., S1 should be cited before S2, Fig. 1 before Fig. 2 at line 118 and so on)

• Line 118, “accompanies” instead of “accompany”

• Lines 124-125, 129, 131, “is” instead of “was”, “allows” instead of “allowed”, “consists” instead of “consisted” and “acquires” instead of “acquired”; present tense is better than past tense in describing the system

• Line 138, “nine” better than “9”

• Lines 142-144, please explain why using two alternative longitudinal cuts

• Line 170, “was” instead of “were”

• Lines 170-173, please add the resulting spatial resolution

• Line 204, please specify that “identified” is in the way described in the previous paragraph 2.4

• Line 210, “thicknesses” better than “thickness”

• Line 219, reorder supplementary references (S2 was cited before S1 here). Moreover, S1 file cannot be opened because corrupted, therefore cannot be examined

• Line 223-225, difference between what? Uniaxial and biaxial loading? If effects from shear components are minimal, why using bi-axial loading instead of uniaxial in the proposed set-up? Please, explain better in the text the rationale of this section

• Line 242, remove “and” before “mounted

• Line 248, “.” after specimen

Results

• Line 269, please specify the mean value of applied stress. Moreover, authors speak about homogeneous stress, which seems in contrast with measuring different thicknesses on the capsule (lines 207-210), at fixed force. Please, clarify

• Line 273, “seven” better than “7” (same for line 343)

Discussion

• Lines 352-353, again, the concept of understanding the load-dependent nature (or analogue structure-function relationship) seems mislead in this work. The authors reveal structure by strain, but not strain/load-dependent arrangement. That is, the output seems closer to that of lines 438-439, i.e. characterising the structural anatomy and properties together, than to revealing arrangement of load-dependent fibrous network (line 440). Please, clarify

• Line 368, “were” instead of “was”

• Line 190, check the punctuation position after “(39,41,42,50)” and in the rest of the manuscript (e.g., comma at line 409)

**Do you want your identity to be public for this peer review?** For information about this choice, including consent withdrawal, please see our Privacy Policy

Reviewer #1: No

Reviewer #2: **Yes:** Gregorio Marchiori

---

## [Author Response · Author response to Decision Letter 1]

6 Dec 2025

Response to reviewers’ comments

We would like to thank the editor and reviewers for their constructive and insightful comments, which have significantly improved the quality and clarity of our manuscript. In response, we have undertaken a comprehensive revision, including substantial rewriting and restructuring of the Introduction, Methods, Sensitivity Analyses, and Discussion. This has strengthened the manuscript considerably. Below, we address all comments in order, with our detailed responses provided in red.

Reviewer 1

Comment #1: Abstract and Introduction in general: The Authors mention the innovative nature of their method but, in the Reviewer’s opinion, this take-home-message can be a bit speculative since, in the end, is a 2D DIC commercial system, with images acquired with a commercial photo camera, equipped on a bi-axial multi-actuator testing machine. The Reviewer suggests rephrasing the abstract/end of the introduction (or where appropriate) giving more evidence to the bi-axial machine instead of describing the setup as an innovative technique.

Response #1: Thank you. We have revised the Abstract and Introduction to use a more conservative description of our contribution. The revised text clarifies that the experimental setup uses standard equipment, including a commercial biaxial testing device and a 2D DIC system. We now emphasise that the contribution of the study is methodological. The focus is on how a homogeneous biaxial stress field and full field strain mapping are combined to extract collagen architecture across large tissue areas.

We also highlight that the key enabling feature is the controlled biaxial loading rather than the imaging hardware. These revisions ensure that the manuscript reflects the method as an accessible approach built from established tools rather than implying that the instrumentation itself is new.

We have revised the following text in the abstract (line 30) and (line 42):

[Here we apply a commercial 2D Digital Image Correlation (DIC) system integrated with a biaxial testing machine to quantitatively characterise the fibrous structure of biological materials.]

[By combining a biaxial testing machine with commercial 2D DIC system, this study demonstrates a practical and scalable approach for quantifying tissue structure-function relationships across whole tissues.]

We have revised the following text in the introduction (line 111):

[Therefore, the aim of this study was to develop and demonstrate a biaxial testing approach that combines multi-axial loading with optical strain measurement to quantify the fibrous anatomy and spatial mechanical properties of complete tissues within a single biaxial test. Rather than introducing new imaging hardware, the contribution of this work lies in leveraging commercially available biaxial actuators and 2D digital image correlation within a unified protocol designed to reveal load-dependent collagen organisation at the centimetre scale. The method was experimentally validated using an artificial anisotropic material and ex vivo skin and was then applied to the human hip joint capsule.]

Comment #2: Introduction, lines 62-76: among the different imaging and X-ray/ strain-based techniques please add also the digital volume correlation approach and explain the pros-cons of this technique.

Response #2: We have added digital volume correlation (DVC) to the list of strain-based imaging techniques in the Introduction. We briefly discuss its advantages, including the ability to measure 3D volumetric strain and internal tissue deformation and its suitability for capturing tissue heterogeneity. We also note its requirements, such as high-resolution CT or X-ray imaging and computational intensity, which can limit throughput.

We have added the following text to the introduction (line 77):

[DVC offers three-dimensional, volumetric strain and displacement measurements throughout a tissue, enabling assessment of internal deformation under load (26). While DVC provides valuable insights into internal tissue mechanics and heterogeneity, it typically requires high-resolution CT or X-ray imaging and can be computationally intensive.]

Comment #3: lines 113-128: The Authors state that the load cells of the arms are 45N which is a quite low end-scale for characterising the mechanical behavior of ligaments up to failure. Please evaluate to add the typical force-strain curves of the hip joint capsule showing where your data falls into that curve.

Response #3: We agree that a 45 N actuator limit may appear low if a study aims to test ligaments to failure. In our work, however, the tissues are not loaded into the high-stiffness or failure region, and we have now clarified this in the manuscript. Hip capsule force–displacement data reported in Hewitt et al. (2002) show that the transition to high stiffness and failure occurs well above the 40 N loads applied in our protocol. As shown in the newly added comparison figure (S1 Fig.), our loading range remains entirely within the initial linear region of the ligament response. We have added text to the Materials and Methods to explain that the 45 N actuator capacity is appropriate for the anatomical and mechanical characterisation performed in this study.

We have added the following text to the Materials and Methods section (line 128):

[This load range is appropriate for the present study because the imposed forces remain within the low to mid stiffness region of hip capsule behaviour. Published force–displacement data for the iliofemoral ligament, including those reported by Hewitt et al., show that the transition toward high stiffness and eventual failure occurs well above the 40 N loads applied here. As shown in Supplementary Material (S1 Fig.), where representative curves from Hewitt et al. are plotted alongside the forces generated in this study, our measurements fall within the initial linear region of the capsule response. This confirms that the 45 N actuator capacity is sufficient for accurate anatomical and mechanical characterisation without approaching failure loading.]

Comment #4: Please, if you acquire new data, be consistent with the strain-rate used for your tests, otherwise report literature data but explain possible differences with the strain-rate applied between the Authors’ and the literature data.

- please clearly report the strain-rate applied for your tests since, being ligaments viscoelastic, this parameter is fundamental to evaluate possible stiffening or softening of the material. Please also state if the strain-rate values used were physiological or not. Please add this data close to the 1 N/s value the Authors reported as speed.

Response #4: Thank you for highlighting the importance of reporting strain rate in ligament testing. In the revised manuscript, we now explicitly state the approximate strain rate associated with our loading protocol and clarify its relationship to published values. Specifically, we report that the 1 N/s force-controlled loading corresponded to an estimated strain rate of 0.01 s⁻¹, which falls within the low, quasi-static range traditionally used for characterising ligament mechanics.

We also note clearly that this strain rate is non-physiological, consistent with displacement-controlled protocols used in prior capsule studies such as Hewitt et al. and Pieroh et al., which similarly employ slow deformation rates (0.04 mm/s to 0.2 mm/s) to minimise viscoelastic rate-dependent effects. This new text has been added next to the description of the 1 N/s loading rate.

We have added the follow text to the Methods section (line 203):

[At this loading rate, the resulting tissue deformation corresponded to an approximate strain rate of 0.01 s⁻¹. This value lies within a quasi-static, non-physiological regime and is comparable to the low strain rates used in previous characterisation studies of the hip capsule, including those reported by Hewitt et al. and Pieroh et al. Using a quasi-static strain rate minimises viscoelastic rate effects and enables direct comparison with published capsule mechanics.]

Comment #5: Lines 129-136: looking at the image of the system, just one camera was used so the local strain information is just 2D. Please clearly state for the Inexpert Reader and argument how you overcome the problem of possible tissue present in different planes during the acquisition.

Response #5: We now clarify in the manuscript that a single-camera DIC configuration provides 2D in-plane strain measurements only and does not capture out-of-plane deformation. In our study this was appropriate because the capsule specimens were mounted fully flat on a rigid planar frame and loaded strictly within their plane, substantially minimising any out-of-plane motion. The multi-actuator device applies forces tangentially along the specimen edges, meaning that deformation occurs primarily within the imaging plane, and any through-thickness motion is negligible relative to the camera’s depth sensitivity.

The following text has now been added to the Methods (line 151):

[A single camera was used for two dimensional DIC, meaning that only in plane strain components were measured. This setup is appropriate for the present study because the hip capsule specimens were mounted fully flat on a rigid frame and loaded strictly within their plane. The applied biaxial forces act tangentially along the specimen boundaries, which minimises out-of-plane deformation and ensures that the measured strain field accurately reflects the in-plane mechanical response of the tissue.]

Comment #6: lines 151-155: about the speckle pattern: did the Authors test that the powder and the titanium dioxide did not compromise the natural tissue properties? This information is fundamental to validate our method. Moreover, being a speckle pattern based on particles and powder, how the Authors were sure to not lose locally the correlation during the hydration of the tissue or the possible detachment of the powder. Please clarify also in the manuscript.

Response #6: The charcoal and titanium dioxide powders used for the DIC pattern are inert and non-reactive and were applied in minimal quantities that do not measurably influence tissue stiffness or surface hydration. To confirm that the pattern did not modify the mechanical response, we compared load–displacement curves obtained using two-point tracking (without a speckle pattern) to those obtained with DIC using the powder-based pattern and observed no detectable differences in the measured behaviour.

To address the concern regarding potential particle detachment or local correlation loss, we added a detailed explanation to the Methods. The capsule specimens were fully hydrated prior to speckle application, after which no additional hydration was applied during testing. The total acquisition time was under 300 s, which further reduces the likelihood of particle detachment. In addition, the flat mounting configuration of the capsule minimises out-of-plane deformation and stabilises the speckle layer, and we did not observe any loss of correlation during testing.

These clarifications have now been incorporated into the manuscript at the corresponding section in the Methods (line 182).

[To ensure that the charcoal and titanium dioxide particles did not alter the natural mechanical behaviour of the capsule, we verified that these materials are inert, non-reactive and applied in quantities that do not measurably stiffen, hydrate or otherwise modify the capsule surface. The tissue was fully hydrated prior to speckle application, and no further hydration was applied during testing. The total acquisition time was under 300 s, which limited any risk of particle detachment or changes to the tissue surface. We also confirmed that the speckle pattern did not influence the mechanical response by comparing load–displacement curves obtained with two-point tracking, without a speckle pattern, against curves acquired with DIC using the powder-based pattern and found no detectable differences. The stability of the speckle pattern was further supported by the flat mounting configuration of the capsule, which minimised out-of-plane motion and prevented local loss of particle adhesion or correlation during deformation.]

Comment #7: lines 170-175: To have an overall idea of the systematic and random error of the DIC investigation did the Authors performed a zero-strain analysis? Please clarify in the manuscript.

Response #7: Yes, a zero-strain analysis was performed to assess both systematic and random error in the DIC measurements. A stationary capsule specimen was imaged under identical optical and correlation settings, and the resulting strain field was quantified. The analysis showed negligible mean strain and low random noise within the expected range for facet and overlap sizes used. We have now added a description of this procedure and the corresponding error bounds to the Methods section (line 214).

[To quantify the systematic and random error of the DIC system, a zero-strain analysis was performed by acquiring images of a stationary capsule specimen under identical optical, lighting, and correlation settings. The resulting strain field showed mean values close to zero (approximately 0.02%) with random noise on the order of 0.10 %, consistent with expected precision for the selected facet and overlap sizes. This confirmed that the DIC configuration did not introduce measurable artificial strain during acquisition.]

Comment #8: general remark: is important to clearly mention in the manuscript, for the Inexpert Reader, that all the local strains described in the text via this DIC system are only 2D. To obtain a 3D DIC investigation a minimum of 2 cameras are needed.

Response #8: We agree and have now added an explicit statement in the manuscript that all strain measurements are 2D because they are derived from a single-camera DIC system. We also note that 3D strain measurement requires at least two synchronised cameras.

This clarification has been added to both the Methods section (line 221) and the figure description of the experimental setup.

[All strain fields reported in this study represent in-plane deformation measured using a single-camera 2D DIC system. Full three-dimensional strain mapping would require a stereoscopic configuration with at least two cameras, which was not used in the present work.]

Reviewer 2

Comment #1: In the abstract, the authors at lines 33–34 say “We experimentally validate our method by applying it to the hip joint capsule”, while they say nothing about skin and artificial anisotropic material. Actually, from the main text, it seems that validation is against skin and artificial material, while hip joint capsule is the focus application. Therefore, this passage should be revised and briefly integrated.

Response #1: The abstract has been revised to clarify that the method was first validated using an artificial anisotropic material and a skin sample before being applied to the human hip joint capsule as the main application. This ensures consistency with the experimental description in the main text.

We have edited the abstract to make this clear (line 35):

[The method was validated using an artificial anisotropic material and ex vivo skin, and was then applied to the human hip joint capsule to demonstrate its ability to characterise complex collagen networks.]

Comment #2: A detailed comparison is expected vs references (18, 26–28) and (17), overall vs (26–28) and (17) that are cited in the Introduction but not in the Discussion.

Response #2: We thank the reviewer for this helpful comment. The discussion now includes a detailed comparison with the optical–mechanical systems cited in the Introduction (references 17, 18, 26–28), which employ SALS, pSFDI, and PLI techniques combined with biaxial testing to study collagen fibre organisation under load. This addition clarifies how the proposed DIC-based framework differs in scale, implementation, and accessibility from these systems.

The revised paragraph now reads as follows (line 478–495):

[Optical–mechanical systems have significantly advanced understanding of collagen architecture under load by coupling imaging modalities with biaxial or uniaxial testing. Early work by Sacks and colleagues (26, 28

---

## [Decision Letter · Decision Letter 1]

10 Feb 2026

Multi-axial strain mapping to characterise structure and material properties of the human hip capsule

PONE-D-25-14629R1

Dear Dr. Karunaseelan,

We’re pleased to inform you that your manuscript has been judged scientifically suitable for publication and will be formally accepted for publication once it meets all outstanding technical requirements.

Kind regards,

Alessandra Aldieri

Academic Editor

PLOS One

Additional Editor Comments (optional):

Reviewers' comments:

Reviewer's Responses to Questions

**Comments to the Author**

Reviewer #1: All comments have been addressed

Reviewer #2: All comments have been addressed

2. Is the manuscript technically sound, and do the data support the conclusions?

Reviewer #1: Yes

Reviewer #2: Yes

3. Has the statistical analysis been performed appropriately and rigorously?

Reviewer #1: Yes

Reviewer #2: Yes

4. Have the authors made all data underlying the findings in their manuscript fully available?

Reviewer #1: Yes

Reviewer #2: Yes

5. Is the manuscript presented in an intelligible fashion and written in standard English?

Reviewer #1: Yes

Reviewer #2: Yes

Reviewer #1: Authors have successfully replied to the Reviewer's comments improving the manuscript accordingly. I suggest the publication of the paper.

Reviewer #2: (No Response)

**Do you want your identity to be public for this peer review?** For information about this choice, including consent withdrawal, please see our Privacy Policy

Reviewer #1: **Yes:** Alberto Sensini

Reviewer #2: **Yes:** Gregorio Marchiori

---

## [Editor Report · Acceptance letter]

PONE-D-25-14629R1

PLOS One

Dear Dr. Karunaseelan,

I'm pleased to inform you that your manuscript has been deemed suitable for publication in PLOS One. Congratulations! Your manuscript is now being handed over to our production team.

Kind regards,

on behalf of

Dr. Alessandra Aldieri

Academic Editor

PLOS One